# A qualitative analysis of the impact of COVID-19 restrictions on gender biases in an Irish University

**Mohammad Hosseini**[1]*, **Alicia Castillo Villanueva**[2]

**1** Department of Preventive Medicine, Northwestern University Feinberg School of Medicine, Chicago, Illinois, United States of America, **2** School of Applied Language and Intercultural Studies (SALIS), Dublin City University, Dublin, Ireland

\* mohammad.hosseini@northwestern.edu

**Data Availability Statement:** The data cannot be shared as this was the the condition set by the Dean of the Faculty where we interviewed

## Abstract

This paper explores the impact of COVID-19 restrictions on gender biases in a STEM Faculty in an Irish University. For the purposes of this research, gender bias was understood as gender-based inclinations or prejudices which affect researchers' personal and professional opportunities as described by fifteen interviewees (nine women, six men). We used thematic interviews to explore participants' perspectives. Analyzing interviews with an inductive approach showed that gender biases experienced before COVID-19 restrictions were different from biases during restrictions. In discussing gender biases prior to the pandemic, interviewees mentioned institutional disparities, discrimination, implicit biases, stereotypes and gender roles. When discussing gender issues during COVID restrictions, interviewees mentioned disparities at home, challenges involved in parenting, support from colleagues and the University, as well as negative and positive impacts of restrictions on existing gender issues. Our results show that while COVID-19 restrictions and the associated consequences constrained all gender groups, they most negatively affected women researchers with caring responsibilities. Moving forward, reducing gender disparities requires promoting a horizontal organizational structure, and adjusting policies and work arrangements to cater for vulnerable cohorts such as parents, and international and early-career researchers. Offering a hybrid working model that grants more flexibility to those with caring responsibilities and which accommodates personal circumstances would help improve the working conditions for all researchers and nurture an organizational culture of care for the employees; thereby also fostering gender equity and tolerance.

## Introduction

### Background

The COVID-19 pandemic and several lockdowns in the Republic of Ireland affected many sectors including education. On 12 March 2020, the Irish government announced the first lockdown and subsequent closure of all childcare facilities, schools and higher education institutions. As the restrictions began, the European Institute for Gender Equality highlighted the gendered effects of restrictions, anticipating that the increased caring responsibilities due

researchers. Additionally, the IRB approval stipulated that the data would not be shared.

**Funding:** This research was supported by a Postgraduate Research Student Journal Publication Scheme Grant [M.H.] and Journal Publication Scheme Grant [A.C.V.] from Dublin City University's Faculty of Humanities & Social Sciences [M.H.] and the Northwestern University Clinical and Translational Sciences Institute (NUCATS, UL1TR001422) [M.H.]. The funders had no role in study design, data collection and analysis, decision to publish, or preparation of the manuscript.

**Competing interests:** The authors have declared that no competing interests exist.

to school closures would negatively affect women [1]. For researchers, lockdown necessitated the sudden transition of all teaching, research and associated support activities to virtual spaces. Recent quantitative analyses suggest that this transition had a more negative effect on women, parents of young children, and people of color [2–5].

Among these groups, the challenges faced by women are frequently explored in the literature. For example, compared to men, women researchers reported undertaking fewer new research projects due to greater childcare and household responsibilities [6–8]. Furthermore, since on average, women faculty staff are more involved in teaching, the transition to online teaching and the required adjustments disproportionately affected them; resulting in the submission of fewer funding applications, recruitment interruption, and uncertainty regarding renewal of temporary contracts [9]. Although the overall negative impact was disproportionately greater on early career researcher, due to social distancing requirements, women in ostensibly powerful positions like Principal Investigators (PIs) were also negatively impacted because of the likelihood of women PI's to have smaller labs which could not accommodate social distancing requirements [7]. While some of these disparities could be attributed to personal preferences (e.g., preferring smaller teams) or institutional limitations (lack of resources), some are symptomatic of or resulted from institutional, structural or interpersonal gender biases [10, 11]. For the purposes of this research, gender biases are understood as gender-based inclinations or prejudices which affect researchers' personal and professional opportunities.

Over the last few years, Ireland has made significant strides in addressing gender issues within higher education, successfully implementing various initiatives and policies at both national and institutional levels [12]. Consequently, major milestones have been achieved. For instance, as a result of adopting a structured and centralized approach, Ireland has consistently outperformed the European average in terms of narrowing the gender pay gap [13]. The Senior Academic Leadership Initiative (SALI) is among best practice national initiatives to address the under-representation of women through awarding forty-five senior academic positions [14]. Ireland has also championed various capacity building initiatives in STEM areas (e.g., "increase the numbers of women pursuing STEM subjects, publishing evaluations of public engagement projects that address gender parity in STEM, ensuring that activities and online content represent gender parity and challenge unconscious bias, and developing a toolkit on unconscious bias for education and public engagement initiatives") [15]. More importantly, these initiatives have also been evaluated and results are shared through the publication of reports such as the National Review of Gender Equality in Irish Higher Education Institutions [16]. Despite major improvements and a proactive attitude from the government and higher education institutions, a recent qualitative study that explored gender biases in an Irish University prior to the pandemic reported that women's opportunities for research and publications were negatively impacted by gendered roles, implicit gender biases, women's high professional standards, negative perceptions of women's expertise and accomplishments, family responsibilities and nationality [12].

Since COVID-19 restrictions significantly changed working conditions (e.g., by forcing research, teaching and administrative tasks to the virtual space), we aimed to shed light on perceived biases before and during the COVID-19 restrictions through the exploration of the lived experiences of respondents from different genders in a STEM faculty in an Irish institution, and to collate researchers' views about how gender biases should be addressed.

## Methods

### Study design and recruitment

We employed thematic interviews to explore participants' perspectives and experiences. Our questionnaire was developed based on the results of a qualitative study of gender disparities in

an Irish University [12] and available literature regarding the impact of COVID-19 restrictions on scholarly work and gender disparities in academia, including that of [2, 17–22]. The questionnaire's design was focused on capturing existing disparities prior to the COVID-19 pandemic and highlighting those that were positively/negatively affected as a result of COVID-19 restrictions. M.H. and A.C.V (authors) jointly developed questions and piloted the questionnaire in two interviews. For pilot interviews, we used our network to recruit one man and one woman researcher from the same University where the final recruitment took place. We subsequently improved the questionnaire (available in S1 File) and secured the approval of Ethics Committee in April 2021 (REF: [UNIVERSITY] REC/2021/056). We also acquired permission from the Dean of a STEM Faculty in the University to interview their staff. An email invitation to participate in the study was sent to all staff by the Faculty's Secretary office on 10 May 2021, and was followed by a reminder on 17 May 2021. This resulted in 15 positive responses from researchers and support staff based in six different schools.

## Interviews

All interviews took place over Zoom between 17 May and 1 June 2021 (during the third wave of the COVID-19 pandemic in Ireland where lockdown measures were still in place). Recordings were stored on the cloud on A.C.V.'s password protected Zoom account. All interviewees (including pilot interviewees) received a €25 gift voucher. On average, each interview lasted 50 minutes and 33 seconds. While both M.H. and A.C.V were present at all interviews, we took turns in leading interviews. Prior to the interviews, participants were sent one link to a Google form to provide voluntary sociodemographic information and another to the digital information sheet and consent form (available in S1 File). All participants signed the consent form prior to the interviews and were asked whether they had any questions at the outset of the interview. At the start of the interview, we introduced ourselves along with our respective pronouns to foster an atmosphere of respect and inclusivity. We clarified our credentials and fields of study (M.H. Research ethics and integrity, A.C.V. Gender studies) and indicated our affiliation. The sociodemographic information of the interviewees is presented in Table 1:

**Table 1. Interviewees' sociodemographic information.**

| Sociodemographic Information | Number (N = 15) |
| --- | --- |
| Woman | 9 |
| Man | 6 |
| LGBTQI+ | 2 |
| Married | 8 |
| Has children | 8 |
| Irish citizen | 9 |
| European citizen (non-Irish) | 5 |
| Non-EU citizen | 1 |
| Average age | 39.2 |
| Average number of years employed at current University | 9.5 |
| Support staff | 2 |
| Postdoctoral researcher | 1 |
| Assistant professor | 9 |
| Associate professor | 4 |

### Analysis of interviews

Recorded interviews were transcribed in July and August 2021 by M.H. and A.C.V and all information that could identify participants or the institution was removed. Using an inductive approach [23], we started the analysis by independently coding three randomly selected interviews. After discussing identified codes, we agreed to use 12 category codes to analyze the remainder of the interviews. Upon analyzing all interviews, certain codes were refined and revised. This led to the development of four new codes and the removal of two codes and overlap reduction (some codes were merged), bringing the total number of codes to 11 (the codebook is available in S1 File). Using the finalized codes, we then analyzed all interviews for a second time. After completing the first draft of the results, we sought feedback from two external reviewers (experts in gender issues) and improved our analysis accordingly. Furthermore, to improve consistency in the terminology used, we adjusted terms indicative of language habits with recommended terminology for discussing gender, i.e., men/women researchers instead of male/female researchers [24].

## Results

In line with the questionnaire's three subsections (i.e., before, during and after the restrictions), the results section is organized with the same structure. It should be noted that gender disparities are not carried through all three sections. The reported themes in each section reflect the interviewees' lived experiences in different timeframes relative to the pandemic. In reporting this study, we followed Consolidated Criteria for Reporting Qualitative Research (COREQ) [25], and present the checklist in S1 File. We also provide interviewees' responses to the Likert-type questionnaire in S1 File. Furthermore, in an attempt to protect participants and reduce the likelihood of reidentification (e.g., by means of combining several quotes from the same person), we removed our code-numbers for interviewees when quoting them.

## Before the COVID-19 restrictions

### Institutional gender disparities

(Instances of gender disparity/equality at the University)

Touching on topics such as balanced gender make-up of promotion committees, equal gender split in new hires, and respect for minority groups, all interviewees asserted that gender inequalities have recently improved overall. However, compared with most European institutions, the rate of change was believed to be slow in this Irish University.

All interviewees cited the gender make-up of their school as an important indicator when discussing gender issues. In our sample, there were men and women interviewees working in schools with a "more-or-less equal" gender make-up, and also interviewees from schools dominated by men or women. All men and women who worked in men-dominated schools considered unbalanced gender make-up an issue that should be addressed. Nevertheless, two women and a man who worked in women-dominated schools noted other disparities including unequal recognition of work, and a greater proportion of men in senior positions with permanent contracts and favorable hiring arrangements (e.g., "somebody is joining us shortly, he doesn't have any previous academic experience but has been appointed as Associate Professor").

The overall approach towards diversity was criticized by three interviewees who maintained that "the University is not having the right impact in terms of ensuring that all people of all gender identities are equally recognized". This was further elaborated by an interviewee who

was critical of the accreditation structure of the Athena SWAN international framework to support gender equality within higher education:

> "The notion that Athena SWAN creates equality is a misnomer because universities only use women and ethnic groups as means to get a silver or gold certificate. That's an afterthought. These new hires are not because we welcome diversity. That is because we need to use women and ethnic minorities for the University to obtain status that isn't on the ground".

Two women referred to the University's approach towards maternity leave and mentioned misconceptions and inaccurate communication regarding its impact:

> "Women are allowed to be absent for maternity leave and can also extend that, but this has a direct knock-on effect on promotion. Women who progress are those who actually continued working when they were on leave. However, maternity leave is promoted as if it has no negative impact, which is disingenuous. The promotion matrix is not normalized for maternity leave".

Two men described the University's strategy to improve the gender make-up of staff as discriminatory. One claimed that since "there is an obvious push to increase women's presence in academia", if men were to apply for a job now, they would be discriminated against. This was also echoed by a man with an unsuccessful promotion application who added that "promotions that are based on equality on a 50–50 basis, reduce the chances of men because there are more men in my discipline".

Respondents from the LGBTQI+ communities reported a lack of institutional support. One reported a fear of coming out and still feeling in the closet: "I feel shy about my gender identity with respect to some individuals, my boss being one of them". This interviewee added that they were unsure the University would support them in cases of harassment. This issue was also stressed by another LGBTQI+ interviewee who revealed:

> "Before COVID, I used to walk to my office in a very different way than the straight way to avoid particular people. Unfortunately, one of those people is a deputy head of the school now".

Moreover, they added that their gender identity is often dismissed in events, because LGBTQI+ communities are largely represented by one specific subgroup.

## Discrimination

(Instances where researchers are subject to unfair treatment by colleagues)

Specific examples of discrimination were mentioned by six women interviewees. These included misattribution of credit ("sometimes in meetings something I say is picked up and re-said without giving me the credit for saying it in the first place"), dismissive attitude ("sometimes I feel like my voice is not being heard because I'm a woman, but it would be heard if I was a man at the same career stage"; "particular researchers approaching young female PhD students in a dismissive manner in front of other people"), bullying ("a male colleague that I work with for charity was bullying me in a series of emails"), and chauvinistic attitude ("external to our school, I experienced the behaviour that I found quite chauvinist towards women"). In one case where a man had witnessed discriminatory behavior, we asked whether he intervened as a bystander, but he admitted:

"I try to keep myself out. I know that some people in the School don't like each other but I try to get on with everyone and be like Switzerland".

The intersection of race and gender was mentioned too. One interviewee advocated for students of color in their school:

"[School] has many black women students, but how they are spoken about is horrendous. Some lecturers say 'black students are difficult or demanding', but a student who asks questions is not difficult. They're the ones that I love".

A man interviewee of European (non-Irish) nationality, on the other hand, noted his difference is actually appreciated:

"I could say that my being [nationality] is seen as 'oh, you're bringing something different to the table', so since I came here, I have been treated differently, people listen to me in a positive way".

## Implicit bias and stereotypes

(Instances where members of a gender group are positively/negatively affected because of implicit biases and stereotypes)

Some men were conscious of implicit biases against women and advocated for them; highlighting that "there are greater expectations on women in terms of their contribution across the work-life-balance". One man noted that society expects women (and not men) "to make sure that the family is running okay and that the children are being looked after". Another was frustrated with men who do not acknowledge these issues:

"I have met men in the University that said there's no such thing as gender inequality, it's all made up. They don't believe it, they show no interest and have no knowledge of it, and even when you present facts, they disagree."

Women also advocated for some men and noted that implicit biases in academia harm men too because "only a certain type of man gets promoted". Another woman asserted that not all men researchers fit into the same category and that some are mistreated by the "alpha males". As she elaborated:

"There is so much emphasis on getting the big grants that give you the big visibility. But whether we like it or not, these favor the alpha males. Although collegiality and good citizenship are appreciated, in practice, somebody else should take up the slack so the alpha male can plough ahead and continue to make the headlines."

One interviewee linked the University's obsession with big grants to a general neoliberal attitude towards collaboration:

"Neoliberalism and individualism are espoused, yet you're asked to collaborate. And while women are encouraged to collaborate and be collegial, some men can be individualistic."

One woman highlighted that some men colleagues do not acknowledge her academic merits:

"...they think that you are in your current position only because gender equality has been implemented. This has been said to me for the job I'm in and for other grants. I was told that the only reason why I got the grant was because they needed a woman, not because I was good."

While noting a similar experience, another woman claimed that reacting to a lack of acknowledgement by man colleagues could jeopardize future opportunities

"Standing up for ourselves is often frowned upon because when it comes to women, what is tolerated and appreciated in this University is conformity: what I would call the old-style Irish women conformity."

A similar point was raised by a woman in relation to promotions:

"You must be 'liked' to progress. But only a certain type of woman is liked. If you do not fit that bill, you wouldn't matter. You must be a stereotypical Irish woman to progress because this University is all about impression management."

One LGBTQI+ interviewee noted implicit biases in hiring have not only made them hide their gender identity, but also forced them to pretend to be heterosexual:

"One reason I have the current job is because when applying for my position, I pushed on heterosexual family values and said my partner is here."

Implicit biases against the LGBTQI+ communities were also mentioned by another interviewee who explained:

"If you are flamboyant, meek and mild, you are accepted but if you are not the sanitized and acceptable face for the heterosexual University environment, you get nowhere."

## Gendered roles

(Instances where researchers conduct specific tasks because of their gender).
When asked about gendered roles, two men said that women are more involved in teaching:

"I don't know any woman in my school who is active in research intensely, but I know a few men who the main part of their job is research, and the teaching is something they do on the side."

Another man concurred:

"We have more women in administrative roles that involve pastoral care or teaching rather than research."

Women, on the other hand, attested that men are disproportionately involved in "health and safety roles" or "roles that do not demand a lot of time". One woman framed this observation as a general attitude, observing:

"Men are not in committees: they don't like it. When it comes to research conveners, it's the women. When it comes to tedious research works and running the programs, it's the women. So, the heavy lifting is done by women."

Most women found themselves conforming to administrative or support roles. Only two women reported taking a proactive attitude against gendered roles, one of whom noted:

"In the past some administrative roles would be given to me. When I started to become aware of that, I would not take meeting notes or be the administrative person. When I didn't do these tasks, they were given to a man."

Others, however, reported conforming to such roles, even in the absence of an explicit demands from their department heads or direct line manager:

"I will be the person to clean the teacups or the coffee mugs if they're left out. I'll end up doing that just because they're there, and that's not delegated to me. I do it naturally."

"Sometimes students come to me or another woman to ask questions. This was not assigned to us but maybe we are more approachable or more likely to feel sorry for them. Or perhaps we are an extension of a maternal figure."

## During the COVID-19 restrictions

### Working from home

(Instances of gender disparity/equality at home)

Of the interviewees who had partners and were working from home, none reported a disparity in terms of using available space at home, apart from one woman who used the only extra room in their house as her home office ("my husband worked from the kitchen"). Nine interviewees (three men, six women) had a designated room to work from, and six (three men, three women) worked from their bedroom or living room. All those who worked from their bedroom or living room claimed it has impacted their mental and physical health. As one explained:

". . .mentally working somewhere separate from your bedroom would make a huge difference, because it is very hard to dissociate your relaxation from working area."

Another interviewee who was living in a shared apartment elaborated that:

". . .no separation of work and home space is mentally uncomfortable, and the lack of proper office furniture is physically uncomfortable."

In general, men and women interviewees felt that the restrictions had a more negative impact on women because their responsibilities at home had increased as a result. However, one woman interviewee (whose husband had to work in-person) disagreed:

"I know that women were left to do a lot and in fairness, I was left doing it too, but I think that was just to do with our jobs so I think it comes down to what kind of job we have."

While some disparities related to domestic tasks were mentioned (e.g., "I made him dinner a bit more"; "I did a bit more childcare than her"), all non-single interviewees felt that they were fully supported by their partner. In reacting to the statement of "I felt supported by my spouse/partner" with Likert-type responses, all who had a partner chose agree or strongly agree.

## Parenting

(Instances where references to parenthood are mentioned)

Parenting and the associated difficulties exacerbated by the COVID-19 restrictions were mentioned by all interviewees to some extent. According to parents who lived with their partner, to the extent that it was possible, the overall distribution of childcare responsibilities was equal throughout the restrictions. The three exceptions were two men and one woman; all with partners who worked in-person. That said, one woman noted that she is (by choice) more attentive to her child:

"We would do childcare completely equally, but when I'm working from home and hear my child, I would immediately attend to it."

While interruptions that hindered concentration were cited by both men and women, attending to children was raised as an issue that particularly extended women's working hours, and in one case, led to illness:

"My husband was working in-person full-time, and I was left with three children at home. My job did not decrease in any way. I didn't get COVID-19, but I got quite sick after one month just from the pure exhaustion of minding the children all day and then working all night."

A single mother also reported:

"With the start of restrictions, I found myself getting up at 5:30, doing some work, then home-schooling my children between 9am and 1pm because the youngest has a learning difficulty and I have to sit there with Zoom calls and do every single thing and cheerlead. And then at 1pm my childminder would come. So, I'd get some more work done, then cook dinner, put the kids to bed, and do some more work. So it was literally work every waking hour."

Furthermore, two women said that restrictions negatively impacted their relationship with their children:

"They see me work all the time. They're told 'don't go near mommy'. They'd be asking 'Are you still working, mommy?' or 'Please stop working, mommy.' What they see is that I don't have time for them."

Another mother agreed:

"I have had my children telling me several times 'I hate your job'. That's exactly what they said. All have said, 'If this is what going to college does to you, we're not going.'"

Two men said that regardless of gender, parenting challenges are not always recognized by colleagues with no childcare experience. One noted:

"I've heard from colleagues that 'Oh sure, it's great, you'd be working at home and you'd be able to sort kids and sit down in your office and work.' That's not the reality. The children's presence makes working extremely difficult."

Another man said: "The mindset of those without children is that childcare is not work". These views were further challenged by two interviewees (both single women without children) who had helped colleagues whose parenting responsibilities limited their bandwidth. They were both unhappy to have been asked to do extra work to help colleagues with children. As one explained:

"It's a personal choice to have children, so nobody else should be paying the price. It's not my problem that my colleague wanted children. I shouldn't be working extra because they need flexibility. This is unfair from a professional perspective."

## Collegial support

(Instances where researchers were/were not supported by colleagues)

Although interviewees held diverse views about support (e.g., "help with teaching", "administrative tasks", "grading" and "emotional support"), responses to Likert-type questions showed that all interviewees except for one woman (who reported being supported by women but not men) felt supported by colleagues of different genders. However, open questions revealed that men were mostly excluded from women's support groups; partly due to lacking initiative and personal preferences. Women described support as "reaching out via the phone" or "casual Zooms where we wouldn't talk about work, but have a coffee and talk about other things". These interactions happened in women-exclusive groups and women who had such support mechanisms considered themselves "lucky". When asked why men were not included in these support circles, one woman said: "They never asked or reached out". This was confirmed by a man who admitted: "I never feel the need for calling my colleagues".

When discussing support, men mostly cited interactions and discussions around work. When asked specifically about receiving emotional support, men highlighted one-on-one interactions with friends working elsewhere. One man reported that he does not expect or want colleagues to provide emotional support and challenged the nature of the question:

"I mean this is a trivial question, I want the normal working support that I have for the last years, nothing else, nothing more."

The only man who had proactively reached out to colleagues to organize regular catch-up meetings said:

"There was a higher proportion of women who showed up than there is across the school. So, more women took the opportunity."

When discussing support received from their head of school, 11 interviewees said that given the circumstances, the management did what they could. However, four interviewees (from three different schools, two led by men, and one led by a woman) were dissatisfied. Those dissatisfied with a woman head of school cited her authoritative approach towards management and lack of attention to feedback in decision-making, which resulted in unnecessary workload for staff. Those who were dissatisfied with men heads of school noted there was a "task-focused" attitude towards communication (e.g., "something has to get done. Let's talk about that, OK, we're finished talking about that. Bye") and delayed enquiring about how staff felt (e.g., "a full year after the start of the pandemic he asked how did you feel in the last year?").

## Support from the University

(Instances where researchers did/did not receive support from the University)

Support received/not received from the University was expressed in different ways and various expectations were highlighted. Respondents mentioned material support (e.g., support to buy home office furniture), individual care (e.g., online therapy), effective communication with students (e.g., informing them about challenges of keeping classes going), inclusive decision-making (e.g., involving lecturers in decisions that affected teaching), and support/work-relief for those with caring responsibilities.

Regardless of gender, there was a consensus among interviewees with caring responsibilities that their circumstances were inadequately acknowledged. Two men and two women who had significant teaching responsibilities stated that despite the restrictions the University pressured staff to maintain teaching standards to near pre-COVID-19 levels. They further said that although restrictions increased workload (e.g., transition to "online teaching"), the University did not lower expectations; "the only things that mattered were the students and to look good to the world". Indeed, while two women specifically highlighted free mindfulness classes as supportive, one woman completely disagreed:

> "As if mindfulness is actually going to do anything for you. I didn't ask for it, don't have time for it and don't want it, but they [the University] offer it because that gets them off the hook for the real duty of care to women with caring responsibilities."

Four men said that COVID-19 restrictions and the subsequent challenges reinforced the University's "company" or "business" vibe, its "neoliberal values", and a hierarchical structure with "worker drones, who are just doing what they're told to do, this time due to an emergency". They asserted that this business-like structure has resulted in a tokenistic/tick-boxing attitude to supporting staff regardless of personal circumstances, such as sending emails with a business tone: "I know you're very stressed, but we cannot offer any practical help"; "a generic statement like 'talk to your manager if you have childcare issues, but hey, you still have to work'"; "the first part of the email stating 'let us know what you need' and then telling us here's this extra work you need to do".

## Negative impacts of restrictions on existing gender biases

(Instances where gender biases were exacerbated because of the COVID-19 restrictions)

In discussing negative impacts of restrictions on existing gender biases, opposing views were raised. Eight respondents (five women, three men) believed that biases against women increased due to caring responsibilities. However, four respondents (three men, one woman) stated they know men with childcare responsibilities too, and thus believed that the COVID-19 restrictions have not necessarily worsened existing gender biases. Three women interviewees felt unable to comment on this issue.

Men and women described negative impacts of moving to online communication in various ways but none of the mentioned issues had specific gendered implications. In other words, according to our interviewees, the transition did not uniquely affect one gender. For instance, women reported the challenges of being heard: "It is very hard to speak up in a public Zoom meeting to get your voice heard"; "on Zoom certain people get a lot of airtime, particularly in big meetings where one person does most of the talking". A man with similar views added: "Zoom has created an ideal environment for those who like to run things top-down". Furthermore, women mentioned the limitations of online conversations as a barrier to involvement in organizational decision making (e.g., "we're not asking those little questions that we would ask

over the desks"; "people are less likely to share unofficial information over Zoom or email"; "it's almost like departmental gossip, it's not gossip, it's just nuggets of information that you would normally get in-person, but they're unavailable on Zoom"). Three men noted the inefficiency of online meetings (e.g., "a three second in-person conversation is now a big operation of checking schedules and trying to arrange a Zoom call and connecting and so on").

## Positive impacts of restrictions on existing gender biases

(Instances where gender biases were improved because of the COVID-19 restrictions)

Three respondents (one man, two women) highlighted that "bullying and micro-aggression were less frequent". One woman said:

> "People are less likely to say outrageous things over email or on Zoom, but I feel like at in-person meetings, sometimes people just get fiery and say things that I know they might regret. On Zoom calls and over email, people are much more measured."

In fact, another woman maintained that restrictions improved perceptions about caring responsibilities:

> "People are more understanding of personal circumstances or responsibilities. We are more accepting that a kid might come on to a Zoom call or that someone with children won't work between 8-10am and 3-5pm because they are on school pick-up duty."

## Post COVID-19 restrictions

(Suggestions to improve gender biases)

When asked about suggestions for adjusting policies or developing strategies to improve gender biases, all interviewees referred to opportunities offered by hybrid working conditions, and the new possibilities to support a more diverse teaching/learning demographic in the University. Nevertheless, there was disagreement about how such arrangements should be implemented. Five interviewees (three women, two men) noted that hybrid working arrangements would only benefit those with caring responsibilities if they are flexible enough to accommodate personal circumstances. This point was also made about equal gender make-up of schools:

> "Women are hired alright. Yet, the workload balance or administrative roles that are frequently placed on women don't change. Policies should be directed towards adjusting workloads for individual situations: e.g., based on who has ill parents, young children or children with disabilities."

By contrast, one interviewee suggested that when drafting new policies for working from home:

> ". . .no gender, no life conditions or number of children should be factored in these policies. Because people usually hide behind such reasons [taking advantage of such policies]."

Four interviewees suggested that academic promotion should equally recognize members of all genders as well as ethnic backgrounds. Furthermore, it was argued that since on average women are more involved in teaching, giving greater weight to teaching experiences in assessing promotions could be a more tangible recognition of lecturers than the current annual

teaching awards offered by the University. A similar argument was made in support of administrative staff:

> "Since most administrators are women, offering them the flexibility to choose working hours or remote working options would improve some inequalities."

Other suggestions included "subsidizing crèche on campus", "offsetting childcare costs to support mothers when attending conferences", "schools to appoint a person in charge of diversity, equity and inclusion issues", "reminders about gender equality" and "drafting and sharing an organizational document to achieve gender equity". Five specific suggestions were offered by LGBTQI+ interviewees as follows:

1. The institutional push to use pronouns (e.g., on Zoom) was criticized since it could render certain members of the community vulnerable (e.g., those who have not come out or prefer not to disclose their gender identity).

2. Making the rainbow flag more visible (i.e., in front of the campus or on the University's website) was considered an effective symbol which improves awareness about zero-tolerance regarding harassment.

3. Policies directed towards addressing LGBTQI+ communities' issues should include advocates from a more diverse group within these communities.

4. Inviting professors or speakers from the LGBTQI+ communities to remind students that regardless of their gender, they can climb the career ladder.

5. The instantiation of "regular compulsory meetings for everyone to learn about gender equity".

## Discussion

By interviewing men and women researchers based in a STEM Faculty in an Irish University, the results of this study provided insights into how cohorts with different gender identities have been affected by the COVID-19 restrictions.

### COVID-19 restrictions affected gender bias and deteriorated working conditions for all researchers, particularly for women and minority groups

The findings of this study demonstrated that some gender biases or "micro-aggressions" [26] were less pronounced during restrictions. Nonetheless, while minority groups did not face aggressive colleagues when going to the office, or women endured snide comments about their success, working conditions generally worsened for all researchers, especially for those with heavy teaching and student pastoral care responsibilities: a group that largely consisted of women. According to our interviewees, while the transition to online environments removed the possibility of being harassed in-person and made communication generally more measured and cautious, it amplified a top-down approach. Subsequently, the deterioration of working conditions negatively influenced men who do not match the stereotypical *alpha male* mold and women who do not pass as a stereotypical *conforming female*.

### COVID-19 restrictions had more negative consequences for women's careers

Consistent with other studies which explored the impact of COVID-19 restrictions on women's research output and funding acquisition, the men and women in our study asserted that

restrictions had a more negative impact on women due to increased teaching, student pastoral care, administrative tasks and more importantly, caring responsibilities: all of which are more often carried out by women.

## The long-term negative impacts of COVID-19 restrictions on women should not be overshadowed by a growing number of women in academia

According to the latest report published by the Higher Education Authority in Ireland [27], there has been an increase in the number of women working at all academic levels. That said, compared to men, fewer women applied for jobs in Irish higher education institutions in 2020: Only 34% of the applicants for lecturer, 29% for senior lecturer, 23% for associate professor and 25% for professor positions were women [27]. This data could be interpreted in several ways; none with overall positive prospects for women in the long run.

1. Since women had to take on extra tasks related to child and family care, fewer women applied for academic positions. This combined with an already precarious situation of women and other underrepresented groups in academia puts them in a vulnerable situation and ultimately threatens a balanced gender make-up in universities. The likely long-term effects are significantly greater for early-career women and those with temporary contracts [5]. This is discussed in more detail in a study on gender and precarious work at higher education institutions in Ireland by O'Keefe and Courtois [28]. The authors argue that temporary employments are not well-respected, offer less gross salary, and enjoy fewer benefits or legal protection. Those with such arrangements (mostly women) are placed in a situation comparable with non-citizenship status.

2. It is possible that the higher education sector has been more appealing to men, or, that overall more men are/feel qualified to apply for these positions. Either way, the higher education sector could lose talented women to other sectors (e.g., the industry). In the long run, this could result in reduced quality of education.

3. One can argue that international and domestic incentives such as the Athena SWAN awards have increased women's representation in academia (Athena Swan is a framework to support and transform gender equality within higher education and research, which recognizes and celebrates good practices towards the advancement of gender equality. Committed institutions adopt ten key principles within their policies, practices, action plans and culture to progress equality in higher education and research and to build capacity for evidence-based equality work [29]). This, combined with top-down affirmative action initiatives like the SALI (Senior Academic Leadership Initiative) has resulted in hiring more women relative to the total number of applicants. However, since there are still significantly more women with temporary contracts [27], this short-lived success could be reversed in the long run (e.g., if women with temporary contracts are hired by the industry).

Regardless of the adopted interpretation, it is clear that equating a recent increase in the number of women in academia with gender equity is conceptually flawed and ignores other nuances.

## The COVID-19 pandemic has contributed to the trend of the neo-liberalization of higher education

If Ireland's neo-liberalization of higher education was aggravated by the global economic crisis of 2008, it has been normalized by the COVID-19 pandemic [5]. The process of neo-liberalization of higher education has also been a topic of interest prior to the pandemic [30, 31].

According to O'Keefe and Courtois, research productivity and leadership "promotes masculine embodiments of success and brilliance and hegemonic masculinity as a norm of behaviour and governance" [28]. Our interviewees highlighted that their institution operates like a business, with a focus on acquiring more funding and impression management. Interviewees underscored that acquiring large funding is highly valued as a task well-done by the *alpha males*, whereas women are expected to fit into the stereotype of the *conforming Irish female*. Due to the pandemic, such neoliberal values were prioritized. This resulted in putting additional pressure on already stretched researchers. Our interviewees noted that despite the increased stress of isolation, childcare responsibilities and home-schooling, the perception of less commuting time (thus working more hours) and also early starts and late finishes, resulted in increased demands in productivity.

### Although parenting in general became more challenging for all parents, women were more negatively affected

Childcare during the pandemic was perhaps the toughest challenge parents had to face when childminding facilities and primary and secondary schools were closed. Both men and women interviewees concurred that in general, COVID-19 restrictions more adversely affected women as their immersion in childcare and household duties grew. The challenges highlighted by our interviewees echoed issues raised in other studies. For instance, interviews with 30 women (in Ireland) revealed that during the pandemic, they suffered higher levels of psychological distress due to a changed dynamic of family-work life, and increased childcare and domestic responsibilities [32]. These findings are consistent with our results regarding the negative effects of the pandemic on parents, especially women researchers, thereby intensifying existing gender issues.

### COVID-19 restrictions challenged traditional notions about masculinity

Our interviews showed that parenting and the distribution of childcare responsibilities during COVID-19 restrictions were not always in line with traditional assumptions (i.e., the mother as the primary caregiver). Three men were the primary child caregivers due to their partner's job and one woman interviewee said her husband cut back his working hours to support her with childcare. Enhanced involvement of men in childcare is explored further in a study about masculinity and COVID-19, wherein Ruxton and Burrell stated that although women are still more involved in childcare and domestic tasks, working remotely enabled men to see and feel the need to increase their share of unpaid care [33]. They speculated that this might improve the division of labor at homes, making fathers more involved in child and home care. Nevertheless, some men remain reluctant to take parental leave which can be seen as "unmanly" and negatively affect their career progression since most men do not take parental leave [33]. This may have contributed to a continuation of uneven distribution of labor in some contexts, as highlighted by Clark and colleagues [32]. Indeed, it is reported that some American fathers were unwilling to cut back working hours [34]. Furthermore, another study reported that during the pandemic, American fathers perceived themselves to be spending more time on domestic and childcare tasks than they actually were [35].

### Short and long-term effects of the COVID-19 restrictions will negatively affect all researchers

In the short-term, most researchers experienced an increase in their workload, change of priorities and agendas (e.g., because labs were closed), and stress and isolation, especially due to

the transition from face-to-face to online work and communication. Single men and women acknowledged feelings of loneliness while those with children or ailing parents felt overburdened with additional care obligations. While women were more inclined to arrange and take part in casual gatherings with co-workers, most men reached out to friends outside of their professional network. The restrictions produced mental and physical discomfort for researchers which may spark or amplify mental-health problems in the long run. In short, for 10 to 15% of people "life will not return to normal" [36].

## This study has limitations due to its small sample size and personal views about one faculty within one University in Ireland

More research should be conducted to explore the long-term effects of the pandemic on specific cohorts. Furthermore, the exploration of current conceptions of masculinity and femininity in Irish higher education institutions deserves more attention. In the same way, more research should be carried out with a focus on LGBTQI+ staff in higher education institutions. Given the rapid growth in the migration of research to online platforms, the long-term effects of these changes on research culture and progress merit further scrutiny. Future research relating to gender could aggregate information about contract types, description of titles and tasks, acquired funding, salary, and other indicators to explore gendered aspects of working conditions in a more comprehensive way.

## Conclusion

This study confirms that while the COVID-19 restrictions and the associated consequences constrained all gender groups in academia, they most negatively affected women researchers with caring responsibilities and amplified already existing gender issues for this cohort. We received mixed responses about the impact of the transition to a virtual environment; for some this offered more flexibility, while for others, it led to isolation and more work. Moving forward, minimizing gender disparities requires adjusting organizational structure and culture, current policies and work arrangements, and allocating funds to support the most negatively affected cohorts. Among other solutions, some of our interviewees suggested a hybrid work model which affords more flexibility to those with caring responsibilities and accommodates personal circumstances. Furthermore, the training and appointment of one researcher in each department to monitor and report concerns about gender and diversity issues was introduced as a solution that could cultivate a tailored culture of equity at a local level. Finally, amongst other suggestions was the possibility of giving more weight to teaching and administrative roles in promotion decisions, and paying specific attention to challenges faced by parents and members of the LGBTQI+ communities.

## Supporting information

**S1 File. Supplemental document includes the information sheet, informed consent form, questionnaire, COREQ checklist, and the results of the Likert-type questionnaire.** (DOCX)

## Author Contributions

**Conceptualization:** Mohammad Hosseini, Alicia Castillo Villanueva.

**Funding acquisition:** Mohammad Hosseini.

**Investigation:** Mohammad Hosseini, Alicia Castillo Villanueva.

**Methodology:** Mohammad Hosseini.

**Project administration:** Mohammad Hosseini, Alicia Castillo Villanueva.

**Supervision:** Mohammad Hosseini.

**Validation:** Alicia Castillo Villanueva.

**Writing – original draft:** Mohammad Hosseini, Alicia Castillo Villanueva.

**Writing – review & editing:** Mohammad Hosseini, Alicia Castillo Villanueva.

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
