## [Decision Letter · Decision Letter 0]

21 Dec 2022

PONE-D-22-26961A Qualitative Analysis of the Impact of COVID-19 Restrictions on Gender Biases in an Irish UniversityPLOS ONE

Dear Dr. Hosseini, Thank you for submitting your manuscript to PLOS ONE. After careful consideration, we feel that it has merit but does not fully meet PLOS ONE’s publication criteria as it currently stands. Therefore, we invite you to submit a revised version of the manuscript that addresses the points raised during the review process.

We look forward to receiving your revised manuscript.

Kind regards,

Md. Tanvir Hossain

Academic Editor

PLOS ONE

Journal Requirements:

"N/A"

 This information should be included in your cover letter; we will change the online submission form on your behalf

Reviewers' comments:

Reviewer's Responses to Questions

**Comments to the Author**

1. Is the manuscript technically sound, and do the data support the conclusions?

Reviewer #1: Yes

Reviewer #2: Yes

2. Has the statistical analysis been performed appropriately and rigorously? 

Reviewer #1: Yes

Reviewer #2: N/A

3. Have the authors made all data underlying the findings in their manuscript fully available?

Reviewer #1: Yes

Reviewer #2: No

4. Is the manuscript presented in an intelligible fashion and written in standard English?

Reviewer #1: Yes

Reviewer #2: Yes

5. Review Comments to the Author

Reviewer #1: The Manuscript is written in an intelligible fashion and in standard English. The Methodology part is clear. Accordingly, there is a matching link between findings and discussion section. Overall, the article is a very interesting one and can be accepted without any major revision.

Reviewer #2: This article is original work providing with strong background information and a solid methodological foundation. Reading the article was my immense pleasure and I found the article suitable for further publication. However, I have few queries and recommendations to improve the quality of the article further.

1. Please provide some direction with elaboration to what ground the data collection tool was developed and what areas it covered. This will help the reader to understand the data collection tool and its overall purpose.

2. In result section, the interviewee quotes were reported without providing much of the background of the informants. I would suggest the authors to provide a data chart of the interview informants providing their background information. I would also suggest to keep the informant's confidentiality in mind while moving through the suggestion I made (if the authors agreed to). Next, I would suggest to use anonymous number to report interviewee, such as Informant 1, informant 2 or using pseudonyms etc. Or the professional designation may be used to report particular data (e.g., a post doctoral researcher said that ...) This will tie the connectivity of the readers with the informants when their information has been reported.

Again I am very pleased to read the article.

6. PLOS authors have the option to publish the peer review history of their article (what does this mean?). If published, this will include your full peer review and any attached files.

Reviewer #1: No

Reviewer #2: No

---

## [Author Response · Author response to Decision Letter 0]

22 Feb 2023

We would like to extend our appreciation to reviewers and the editor for their feedback. These have enhanced the overall quality and clarity of this paper.

---

## [Decision Letter · Decision Letter 1]

26 Apr 2023

PONE-D-22-26961R1A Qualitative Analysis of the Impact of COVID-19 Restrictions on Gender Biases in an Irish UniversityPLOS ONE

Dear Dr. Hosseini,

Thank you for submitting your manuscript to PLOS ONE. After careful consideration, we feel that it has merit but does not fully meet PLOS ONE’s publication criteria as it currently stands. Therefore, we invite you to submit a revised version of the manuscript that addresses the points raised during the review process.

Unfortunately, the academic editor who originally handled your manuscript became unavailable, so it has been assessed by our in-house editorial team. We felt that additional reviewer input was required to ensure that your manuscript had received a comprehensive assessment, and hence we have consulted three additional expert reviewers, whose reports are appended to this letter. As you will see from the full comments, two reviewers were broadly satisfied with the manuscript as it stands, while a third feels that there are remaining issues to be addressed relating to how the study fits within the existing literature and the reproducibility of the methodology as currently reported. We apologise that these concerns were not identified in the first round of review. Please ensure you respond to each point raised by all reviewers carefully in your response to reviewers document, and modify your manuscript accordingly. 

We look forward to receiving your revised manuscript.

Kind regards,

Dr Joseph Donlan

Senior Editor

PLOS ONE

Reviewers' comments:

Reviewer's Responses to Questions

**Comments to the Author**

1. If the authors have adequately addressed your comments raised in a previous round of review and you feel that this manuscript is now acceptable for publication, you may indicate that here to bypass the “Comments to the Author” section, enter your conflict of interest statement in the “Confidential to Editor” section, and submit your "Accept" recommendation.

Reviewer #2: All comments have been addressed

Reviewer #3: All comments have been addressed

Reviewer #4: All comments have been addressed

Reviewer #5: (No Response)

2. Is the manuscript technically sound, and do the data support the conclusions?

Reviewer #2: Yes

Reviewer #3: Yes

Reviewer #4: Yes

Reviewer #5: Partly

3. Has the statistical analysis been performed appropriately and rigorously? 

Reviewer #2: N/A

Reviewer #3: Yes

Reviewer #4: N/A

Reviewer #5: N/A

4. Have the authors made all data underlying the findings in their manuscript fully available?

Reviewer #2: No

Reviewer #3: Yes

Reviewer #4: Yes

Reviewer #5: Yes

5. Is the manuscript presented in an intelligible fashion and written in standard English?

Reviewer #2: Yes

Reviewer #3: Yes

Reviewer #4: Yes

Reviewer #5: No

6. Review Comments to the Author

Reviewer #2: The previous issues have been addressed by the author(s) precisely and the paper looks better now. I suggest publishing the paper without further revision from my point of view.

Reviewer #3: Authors have followed guidelines for conducting this qualitative study on very important issue Impact of COVID-19 Restrictions on Gender Biases. and written manuscript following manuscript writing guidelines for qualitative study. It would have been better if authors document limitations of the study.

Reviewer #4: The question is original and well defined. Understanding the consequences of COVID-19 on women and men in STEM in academia, whether directly or indirectly, is important in the recruitment and retention of employees to academia in STEM fields overall. I was happy to see this research being added to the topic of women in STEM, especially a study from outside the United States. The interview responses support the growing body of research regarding the barriers and biases women in STEM in higher education faced during the pandemic. The insights on the LGBTQIA community are sorely needed and much appreciated.

There is one confusing point with Table 1 where the number of participants do not match the employee titles. Support staff, postdoc, assistant/associate = 16; number of participants = 15. Did one participant have dual titles? This was unclear in the table or text.

Reviewer #5: The article presents a qualitative small-scale study on perceptions of the impact of COVID-restrictions on gender bias in an Irish University STEM Faculty based on semi-structured interviews with 15 academic women and men. The authors conclude that gender biases experienced before COVID-19 restrictions were “different from biases during restrictions” (abstract), and that women researchers with caring responsibilities have been mostly affected by the COVID-restrictions. They also present future visions on how to reduce gender disparities, including “promoting horizontal organisational structure” (abstract), adjusting policies and work arrangements to support vulnerable groups, and introducing hybrid working models.

The main weakness of the article is that it is rather thinly relating the findings and discussion of results and future visions to extant and extensive literature on gender inequalities in academia, including much recent published research focusing on Irish academia (e.g. Drew, E., & Canavan, S. (2020). The Gender-sensitive University: A Contradiction in Terms?. Taylor & Francis). There is a robust research base on gender inequalities in academia before the pandemic, and it is unlikely that these inequalities would vanish during the pandemic, rather exacerbate in different ways. Furthermore, thinking of international readers who may not know the Irish context, the authors fail to characterize the Irish national background: the pro-active gender equality approach of Irish higher education with very recent national reviews (2022), the more so relevant given that Ireland belongs to the EU countries which during the recent decade have made notable advances in national policy in this area (see, e.g. EU She Figures 2022). The article needs major revision in these respects, including specifying better when to use which gender concepts (gender equality/equity, gender balance, gender disparities, gender bias, gendered roles). Gender equality/equity is more than gender balance, for example.

Furthermore, the authors do not inform what kind of gender equality structures and policies are in place in the case university. This would also be an important contextual factor when discussing recommendations for the future.

2. Method. The authors do not spell out in the beginning of the methodology subchapter what method is used for data collection but speak about “questionnaire” which is a tool, not a method and could refer to many things. It comes clear only later when reading ahead and consulting additional material, that the data collection method is thematic interview.

The sampling was purposive, and the invitation for interview was sent to all academic and support staff within the chosen Faculty. The authors received only 15 responses and considered it satisfactory, but they do not tell how many invitations were sent in total, and how many women/men were in the staff overall? These details should be explained to be able to assess the consequences for the sampling – was it a biased sample and into what direction?

The authors present necessary technical details on the interviews in the supplementary document. Some of that data could be lifted to the article text to make the methodology subchapter richer. I was wondering how the authors introduced themselves in the interviews – this is rather relevant in this kind of study where academics study academics and does influence trust and informants’ willingness to disclose. What I also missed was some characterisation on the nature and flow of the interviews concerning the ambience of the interviews: hesitations, short answers, emotions?

The socio-economic data table 1: how many identified to belong into LGTBQI+ at least two according to the results part – would be relevant concerning comments on these issues in reporting. Why married a category but not cohabiting? May be more relevant to know how many were single than formally married.

5. The article is generally well and clearly written but the authors should check for typos.

Additional comments:

The meaning of Athena Swan accreditation in the Irish HEI context needs to be explained to international readers, maybe in a footnote.

Figure 1. It is not clear what is the relevance of this figure beyond mere illustrating the topic.

Claim that LGTBQI issues represented “by one specific subgroup” – could the authors tell which? P.10

Gendered roles subheading (p.12) – role terminology may not be the best option here: there is much extant literature on gendered division of labour in academia, and for some aspects of results the concept “academic housework” that is gendered would be more relevant than using the role concept.

p.19. Negative impacts on gender biases by Covid restrictions – no gendered implications? I read the quotes somewhat differently - women reporting more difficulties getting one’s voice heard, and one man commenting more general level suggests gendered patterns – small sample though.

In Discussion: The statement “While the transition to online environments

removed the possibility of being harassed and made communication more measured

and cautious, it amplified a top-down approach.“ This is not a correct claim. According to much recent research, online environments are far from being free from harassment.

7. PLOS authors have the option to publish the peer review history of their article (what does this mean?). If published, this will include your full peer review and any attached files.

Reviewer #2: No

Reviewer #3: **Yes: **RANO MAL Piryani

Reviewer #4: No

Reviewer #5: No

---

## [Author Response · Author response to Decision Letter 1]

9 Jun 2023

Reviewer #2

The previous issues have been addressed by the author(s) precisely and the paper looks better now. I suggest publishing the paper without further revision from my point of view.

Thanks for your feedback.

Reviewer #3

Authors have followed guidelines for conducting this qualitative study on very important issue Impact of COVID-19 Restrictions on Gender Biases. and written manuscript following manuscript writing guidelines for qualitative study. It would have been better if authors document limitations of the study.

Limitations are mentioned at the end of the discussion section.

Reviewer #4

The question is original and well defined. Understanding the consequences of COVID-19 on women and men in STEM in academia, whether directly or indirectly, is important in the recruitment and retention of employees to academia in STEM fields overall. I was happy to see this research being added to the topic of women in STEM, especially a study from outside the United States. The interview responses support the growing body of research regarding the barriers and biases women in STEM in higher education faced during the pandemic. The insights on the LGBTQIA community are sorely needed and much appreciated.

Thanks for your feedback.

Reviewer #5

The article presents a qualitative small-scale study on perceptions of the impact of COVID-restrictions on gender bias in an Irish University STEM Faculty based on semi-structured interviews with 15 academic women and men. The authors conclude that gender biases experienced before COVID-19 restrictions were “different from biases during restrictions” (abstract), and that women researchers with caring responsibilities have been mostly affected by the COVID-restrictions. They also present future visions on how to reduce gender disparities, including “promoting horizontal organisational structure” (abstract), adjusting policies and work arrangements to support vulnerable groups, and introducing hybrid working models.

The main weakness of the article is that it is rather thinly relating the findings and discussion of results and future visions to extant and extensive literature on gender inequalities in academia, including much recent published research focusing on Irish academia (e.g. Drew, E., & Canavan, S. (2020). The Gender-sensitive University: A Contradiction in Terms?. Taylor & Francis). Furthermore, thinking of international readers who may not know the Irish context, the authors fail to characterize the Irish national background: the pro-active gender equality approach of Irish higher education with very recent national reviews (2022), the more so relevant given that Ireland belongs to the EU countries which during the recent decade have made notable advances in national policy in this area (see, e.g. EU She Figures 2022). The article needs major revision in these respects, including specifying better when to use which gender concepts (gender equality/equity, gender balance, gender disparities, gender bias, gendered roles). Gender equality/equity is more than gender balance, for example. 

Thanks so much for sharing these suggestions. We added some more information about the Irish context and highlighted recent achievements. Also, thanks for introducing sources, we engaged with them and cited relevant pieces which were very helpful.

Regarding the used terminology, we’re fully on board with your feedback. We have offered our working definition in the introduction and the abstract “For the purposes of this research, gender biases are understood as gender-based inclinations or prejudices which affect researchers’ personal and professional opportunities.” Thanks to your comment, we revised instances where “gender balance” was mentioned vaguely to highlight what we mean, which is balance in gender make-up. If you have other suggestions please let us know.

2. Method. The authors do not spell out in the beginning of the methodology subchapter what method is used for data collection but speak about “questionnaire” which is a tool, not a method and could refer to many things. It comes clear only later when reading ahead and consulting additional material, that the data collection method is thematic interview. 

You are absolutely right. We added a new sentence to both the introduction and the abstract to clarify. 

The sampling was purposive, and the invitation for interview was sent to all academic and support staff within the chosen Faculty. The authors received only 15 responses and considered it satisfactory, but they do not tell how many invitations were sent in total, and how many women/men were in the staff overall? These details should be explained to be able to assess the consequences for the sampling – was it a biased sample and into what direction? 

We highlight in the manuscript that “An email invitation to participate in the study was sent to all staff by the Faculty’s Secretary office on 10 May 2021, and was followed by a reminder on 17 May 2021”. 

To maintain confidentiality and avoid potentially compromising the institution involved, we are unable to provide any details regarding the size, number of staff or the gender make up of the faculty. We tried our best to provide context through indicating whether a school is dominated by men or women. 

The authors present necessary technical details on the interviews in the supplementary document. Some of that data could be lifted to the article text to make the methodology subchapter richer. I was wondering how the authors introduced themselves in the interviews – this is rather relevant in this kind of study where academics study academics and does influence trust and informants’ willingness to disclose. What I also missed was some characterisation on the nature and flow of the interviews concerning the ambience of the interviews: hesitations, short answers, emotions? 

We added some more details to the methods section. While investigating the flow or ambience of the interviews seems like a great idea, these were not explored in this study.

The socio-economic data table 1: how many identified to belong into LGTBQI+ at least two according to the results part – would be relevant concerning comments on these issues in reporting. Why married a category but not cohabiting? May be more relevant to know how many were single than formally married. 

We added LGBTQI+ to our table. While we did not specifically ask about cohabitation, none of our single interviewees were cohabiting.

5. The article is generally well and clearly written but the authors should check for typos. 

We had the piece professionally proofread and to the best of our knowledge, there should be no typos. If you see any, please let us know.

Additional comments: 

The meaning of Athena Swan accreditation in the Irish HEI context needs to be explained to international readers. 

We added a few sentences to clarify.

Figure 1. It is not clear what is the relevance of this figure beyond mere illustrating the topic.

We deleted the figure.

Claim that LGTBQI issues represented “by one specific subgroup” – could the authors tell which? P.10

We cannot disclose this because it could compromise the identity of the interviewee.

Gendered roles subheading (p.12) – role terminology may not be the best option here: there is much extant literature on gendered division of labour in academia, and for some aspects of results the concept “academic housework” that is gendered would be more relevant than using the role concept. 

Although we agree that administrative tasks could be seen as academic housework, we respectfully disagree with the suggestion to replace the terminology because we explored this in the context of what is asked of women as well as men and so we believe that “Gendered roles” is more appropriate here. For example, “health and safety roles” that are often assigned to men or “teaching” which was believed to be assigned to more women are not necessarily housework. 

p.19. Negative impacts on gender biases by Covid restrictions – no gendered implications? I read the quotes somewhat differently - women reporting more difficulties getting one’s voice heard, and one man commenting more general level suggests gendered patterns – small sample though. 

Our emphasis here was on existing biases and issues that were felt more by one gender. For example “not being heard” was also highlighted by a man. Thanks to your feedback, we added a sentence to clarify.

In Discussion: The statement “While the transition to online environments

removed the possibility of being harassed and made communication more measured

and cautious, it amplified a top-down approach.“ This is not a correct claim. According to much recent research, online environments are far from being free from harassment.

Thanks for bringing this to our attention. We meant that the possibility of in-person harassment was removed and clarified this in the paper.

---

## [Decision Letter · Decision Letter 2]

28 Jun 2023

A Qualitative Analysis of the Impact of COVID-19 Restrictions on Gender Biases in an Irish University

PONE-D-22-26961R2

Dear Dr. Hosseini,

We’re pleased to inform you that your manuscript has been judged scientifically suitable for publication and will be formally accepted for publication once it meets all outstanding technical requirements.

Kind regards,

Sylvester Chidi Chima, M.D., L.L.M, LLD.

Academic Editor

PLOS ONE

Additional Editor Comments (optional):

Reviewers' comments:

Reviewer's Responses to Questions

**Comments to the Author**

1. If the authors have adequately addressed your comments raised in a previous round of review and you feel that this manuscript is now acceptable for publication, you may indicate that here to bypass the “Comments to the Author” section, enter your conflict of interest statement in the “Confidential to Editor” section, and submit your "Accept" recommendation.

Reviewer #3: All comments have been addressed

Reviewer #4: All comments have been addressed

2. Is the manuscript technically sound, and do the data support the conclusions?

Reviewer #3: Yes

Reviewer #4: Yes

3. Has the statistical analysis been performed appropriately and rigorously? 

Reviewer #3: Yes

Reviewer #4: Yes

4. Have the authors made all data underlying the findings in their manuscript fully available?

Reviewer #3: Yes

Reviewer #4: Yes

5. Is the manuscript presented in an intelligible fashion and written in standard English?

Reviewer #3: Yes

Reviewer #4: Yes

6. Review Comments to the Author

Reviewer #3: Authors have followed guidelines for conducting this qualitative study on very important issue Impact of COVID-19

Restrictions on Gender Biases, and written manuscript following manuscript writing guidelines for qualitative study

Reviewer #4: (No Response)

7. PLOS authors have the option to publish the peer review history of their article (what does this mean?). If published, this will include your full peer review and any attached files.

Reviewer #3: **Yes: **Rano Mal Piryani

Reviewer #4: No

---

## [Editor Report · Acceptance letter]

19 Sep 2023

PONE-D-22-26961R2 

A Qualitative Analysis of the Impact of COVID-19 Restrictions on Gender Biases in an Irish University 

Dear Dr. Hosseini:

I'm pleased to inform you that your manuscript has been deemed suitable for publication in PLOS ONE. Congratulations! Your manuscript is now with our production department. 

Kind regards, 

on behalf of

Professor Sylvester Chidi Chima 

Academic Editor

PLOS ONE